# A Highly Accurate Method for Measuring Response Time of MEMS Thermopiles

**DOI:** 10.3390/mi13101717

**Published:** 2022-10-11

**Authors:** Zeqing Xiang, Meng Shi, Na Zhou, Chenchen Zhang, Xuefeng Ding, Yue Ni, Dapeng Chen, Haiyang Mao

**Affiliations:** 1University of Chinese Academy of Sciences (UCAS), Beijing 100049, China; 2Wuxi IoT Innovation Center Co., Ltd., Wuxi 214029, China; 3Institute of Microelectronics of Chinese Academy of Sciences, Beijing 100029, China; 4Jiangsu Hinovaic Technologies Co., Ltd., Wuxi 214135, China

**Keywords:** response time, microheater, MEMS, thermopiles

## Abstract

The response time is an important parameter for thermopiles sensors, which reflects the response speed of the device. The accurate measurement of response time is extremely important to evaluate device characteristics for using them in suitable scenarios. In this work, to accurately measure the response time of thermopile sensors, an Al microheater is integrated in a MEMS thermopile as an in situ heat source. Compared with the traditional chopper measurement method for response time, this approach avoids mechanical delay induced by chopper blades. Accordingly, based on this approach, the response time of the device is measured to be 6.9 ms, while that is 12.7 ms when a chopping system is used, demonstrating that an error of at least 5.8 ms is avoided. Such an approach is quite simple to realize and provides a novel route to accurately measure the response time.

## 1. Introduction

MEMS thermopiles can convert infrared radiation into electrical signals and have been widely used in non—contact thermometers [1,2], uncooled infrared cameras [3,4], gas flow sensors [5,6,7], heat flow sensors [8,9,10], nondispersive infrared sensors [11,12,13,14,15], and vacuum gauges [16,17], etc. The performance of thermopiles can be evaluated by various parameters, including responsivity, detectivity, response time, etc. Herein, the response time is the time required by the device to detect the object and reach the stable state, which reflects the response speed of the device to external excitation [18,19]. In the past few decades, previous research has been mainly focused on shortening the response time of the device by optimizing thermopile structures [20,21,22,23,24]. However, accurate measurement methods for this parameter have not been well developed. It should be noted that the more precisely the response time of the device can be measured, the more useful the device will be in various applications. Accordingly, to obtain an accurate response time for the thermopile sensors, a high—precision measurement system is required.

Traditionally, the test of response time requires a complex system involving a blackbody radiation source, a chopper, and other equipment. With rotation of the chopper blades, the test system provides a changing radiation so that the sensor outputs the corresponding voltage, from which the response time of the device can be obtained [20,22,23,24,25,26,27,28]. Nevertheless, it takes time for the chopper blades to rotate, which prolongs the response time of the device. Besides, after the chopper is used for a long time, the mechanical wear of the blades may also concomitantly increase the measurement error of the response time. Since the delay caused by the chopper cannot be excluded from the test system, there is an inevitable error in the response time by using the traditional measurement system. 

To avoid such an error, Zhang et al. adopted a pulsed laser—based system for response time measurement [19]. In that work, a pulsed laser is used to measure the response time of the thermopiles. Though the system error is in the ps level due to the high precision of the laser equipment, the method is complex and low operable, so that it is easy to cause the measurement error when the sensor sample is changed every time, and the laser may irradiate on different position of the thermopile in the measurement. Therefore, it is necessary to develop a simple, stable, and accurate method for measuring the response time of thermopile sensors.

In this work, a novel approach is designed to accurately measure the response time of MEMS thermopiles through in situ integration of an Al microheater in the sensor. Using the advantages of a short distance from the Al microheater to the hot ends and the corresponding extreme—short heat transfer time, the measurement response time with a very small error is realized. Besides, on—chip integration of the heat source with the thermopile is quite simple and easy to fabricate. It is expected that such an approach has wider application prospects as it can be extended to other MEMS thermal sensors in the future.

## 2. Design and Working Principle

### 2.1. Response Time of Thermopile Sensors

According to the Seebeck effect, the output voltage of a MEMS thermopile can be described as:(1)$$V_{\mathrm{out}}=N∆T\left(\alpha_{A}-\alpha_{B}\right)$$
where $\alpha_{A}$ and $\alpha_{B}$ are the Seebeck coefficients of thermocouple materials A and B, respectively. ∆T is the temperature difference between the hot and cold ends of the thermocouples, and N is the number of the thermocouples.

Besides, the response time of a sensor refers to the time required for the output voltage to reach 63% of its stable value [19], which is defined by the following formula:(2)$$V_{t}=0.63V_{\mathrm{out}}=0.63\;\left(V_{\max}-V_{\min}\right)$$
where $V_{\mathrm{out}}$ is the stable output voltage of the thermopile device, $V_{t}$ is the output response voltage. From $V_{t}$, it is deduced that t is the response time of the thermopile device. In the presence of external excitation, $V_{\max}$ is the maximum value of output voltage, and $V_{\min}$ is the minimum value.

### 2.2. Structure Design of the Thermopile Sensor

In order to obtain the response time of the device with high accuracy, a microheater is embedded around the hot ends of the thermocouples in a MEMS thermopile. When a voltage is applied to the microheater, the temperature on and around the microheater rises, thus will create a temperature difference between the hot and cold ends of the thermocouples. Figure 1 shows the working principle diagram of the approach.

Based on this working principle, a novel structure of the MEMS thermopile is designed, as shown in Figure 2. The sensor consists mainly of a silicon substrate, a supporting layer, stacked thermocouples composed of N—PolySi and P—PolySi strips, a microheater, and a light absorber. The supporting layer uses a “sandwich” structure consisting of SiO_2_, Si_3_N_4_, and SiO_2_ to decrease the tensile stress. The light absorber of a Si_3_N_4_ layer is located at the top level. The N/P PolySi strips are connected with metal forming a series of thermocouples. 

In order to transfer the heat energy generated by the microheater to the thermocouples more accurately, a microheater is designed to be embedded around and as close as possible to the hot ends of thermocouple strips. Since the solid thermal conduct occur between the microheater and the thermopile, the reduction of the distance between the microheater and the thermopile will shorten the heat transfer time to the thermopile when the microheater acts as a heat source. Considering the manufacturing capability of our laboratory, a 3 µm interval is selected between the microheater and the thermopile during the fabrication process, as depicted in Table 1. Besides, to ensure that each pair of thermocouple strips can receive heat energy from the microheater and reduce the complexity of the fabrication process, the microheater is designed to be embedded between the hot ends of every two thermocouple strips. In the fabrication process, the metal connection lines, electrodes, and the microheater are formed simultaneously.

The key structural parameters of the thermopile are shown in Table 1. The microscopic image of the device is displayed in Figure 3. By applying an input voltage for the microheater to generate joule heat, the heat transfers to the hot ends of the thermocouple strips through solid heat conduction. Subsequently, a temperature difference between the hot and the cold ends is generated, then, an output voltage is generated in the sensor, which can be detected from the electrodes of the thermopile. According to Equation (2), the output response voltage and the response time of the thermopile can be obtained.

## 3. Results and Discussion

### 3.1. Effects of Microheater Materials

For a typical microheater, the relationship between its resistance and temperature can be expressed by the following equation:(3)$$R\left(T\right)=R\left(T_{0}\right)\left[1+m\left(T-T_{0}\right)\right]$$

Here R(T) is the resistance of the microheater when its temperature reaches *T*, m is its temperature coefficient of resistance (TCR), $R\left(T_{0}\right)$ is the resistance of the microheater at room temperature $T_{0}$, which can be expressed as:(4)$$R\left(T_{0}\right)=\eta \frac{L}{S}$$

Here η is the resistivity of the microheater material, L is the length of the microheater, and S is the cross—sectional area of the microheater.

To analyze the resistance of the microheater, according to Table 2 (resistivity, η (Ω.m); TCR, *m* (10^−4^/K)) as well as Equations (3) and (4), the resistance is proportional to the resistivity and TCR. Since the input pulse voltage generated by the signal generator is a constant voltage signal, the joule heat from the microheater is inversely proportional to the resistance. In order to obtain more joule heat, Al, Cu, and Ag are analyzed and compared, as these materials all have low resistivity and small TCR.

With the microheater working, the response time of the thermopile can be expressed as [23]:(5)$$t=\left(\frac{1}{N}\cdot \overset{5}{\underset{i=1}{\sum }}\frac{l_{i}}{\lambda_{i}\cdot d_{i}\cdot w_{i}}\right)\left[\overset{5}{\underset{i=1}{\sum }}l_{i}\cdot w_{i}\cdot d_{i}\cdot \rho_{i}\cdot c_{i}+A_{d}\cdot \overset{7}{\underset{i=6}{\sum }}(d_{i}\cdot \rho_{i}\cdot c_{i})\right]$$
where l, w, d, λ, ρ, c denote the length, width, thickness, thermal conductivity, mass density and heat capacity of each part of the thermopile zone, respectively (i = 1, the parameters of P—PolySi thermocouple strips; i = 2, the parameters of N—PolySi thermocouple strips; i = 3, the parameters of the thermal insulation layer of SiO_2_; i = 4, the parameters of dielectric support layer; i = 5, the parameters of microheater; i = 6, the parameters of the absorber; i = 7, the parameters of supporting membrane of the absorber; $A_{d}$, the area of the infrared radiation absorber).

By analyzing the response time of the thermopile device when the microheater was working, it can be known from Equation (5) and Table 2 (mass density, ρ (kg/m^3^); heat capacity, c (J/(kg.K)); thermal conductivity, λ (W/(m.K))) that the response time was proportional to the mass density and heat capacity, and inversely proportional to the thermal conductivity [29]. In other words, the response time is proportional to the ratio of the product of the material’s mass density and heat capacity to thermal conductivity (RMHT). Considering these three factors, it can be seen from Table 2 that the RMHT of Al, Cu, and Au was relatively small; thus, they are suitable for preparing the microheater.

In summary, comparing with the changes of the microheater resistance caused by the different resistivity and TCR of different materials and the changes in the response time and output voltage generated by the working microheater, it is deduced that Al and Cu are more suitable for constructing microheaters. As Al is compatible with the CMOS process and easier to be patterned in the manufacturing process, therefore, Al is adopted as the material of microheater in this work. To further verify the conception, the properties of Pt and PolySi are compared and analyzed with those of Al in the following work, since previous studies mostly used Pt and PolySi as the microheater materials [5,27,30,31].

### 3.2. Simulation of Response Time

From Table 2 and Equation (5), it can be seen that the RMHT of PolySi is quite large, which will increase the response time of the device. The resistance of the microheater is relatively large when a stable pulse voltage is applied to the PolySi microheater, which will severely reduce the thermal energy of the microheater, resulting in a very low voltage of the thermopile. Therefore, PolySi is not discussed according to these theories. In order to ensure that the device can obtain greater thermal energy and reduce the influence on response time, the thermal distribution and the device output with a microheater consisting of Al and Pt are simulated, respectively. As depicted in Figure 4, when applied with a 5 V heating voltage, the output voltages of the device with Al and Pt microheaters are 2.58 V and 0.73 V, respectively. Besides, the response times of the device are simulated to be 7.4 ms and 8.7 ms, respectively, as shown in Figure 5.

From the results, it could be clearly seen that compared with a Pt microheater, the Al microheater can enhance the output voltage of the thermopile device by 253% while reducing the response time by 15%. This is because the resistance of Al is much smaller than that of Pt, as well as the RMHT of the Al is also much smaller than that of Pt. From Equations (3) and (5), it can be deduced that the output voltage and the response time vary accordingly. Based on these reasons, Al is selected as the microheater material in thermopile fabrication.

### 3.3. Measurement System for Response Time

#### 3.3.1. A Chopper—Based System

As shown in Figure 6a, a chopper—based system for measuring response time consists of a blackbody radiation source, a mechanical chopper, a thermostat, and a semiconductor parameter analyzer, etc. During the measurement, the device was placed in a thermostat, which was used to keep the detector at an ambient temperature of 23 °C consistently. The temperature of the blackbody was set at 500 K. With the function of the chopper, the infrared radiation was modulated into a 5 Hz alternating signal before reaching the device in the thermostat. Subsequently, an alternating voltage signal with the same frequency was exported by the detector and displayed on the semiconductor parameter analyzer after passing through a low—pass filter circuit module. As illustrated in Figure 6b,c, the response time obtained by the chopper—based system was 12.7 ms.

#### 3.3.2. A Microheater—Based System

Figure 7a illustrates an in situ microheater—based system, which consists of a signal generator, a thermopile sensor embedded with a microheater, a smoothing filter circuit module, and an oscilloscope. In the measurement, a square wave voltage of 5 V @ 5 Hz generated by the signal generator was applied to the microheater, then the output voltage of the thermopile was captured by the oscilloscope after being filtered by a smoothing filter circuit module. As depicted in Figure 7 b,c, the response time based on the microheater system was 6.9 ms.

By comparing the response time obtained from the two systems, a difference of 5.8 ms is observed, indicating that the microheater—based approach has a 46.7% improvement in the accuracy of response time than the chopper—based method. The reason is that the shading and light—transmitting parts in the chopper have specific and fixed areas; delayed periods are induced when the chopper blades are rotating, which also becomes a part of the response time of the device. Besides, In the long—term test of the chopper, the blades of the chopper may wear out or drift in rotational speed, which will lead to problems of inconsistent sampling frequency and longer sampling time. However, the measurement system with an in situ microheater can avoid the delay caused by the chopper. Table 3 displays the series of chopper—based test systems and the microheater—based system compared by response time parameters. It is seen that compared with the chopper—based system, the response time measurement of the microheater—based system to the thermopile sensor was more accurate.

## 4. Conclusions

In this work, a MEMS thermopile sensor with an in situ integrated microheater between the thermocouple’s strips is designed and fabricated. In the device, Al was chosen as the material for the microheater. Compared with the thermopile of the Pt—based microheater, the output voltage of the Al—based device can be increased by 253% in performance, and the response time can be decreased by 15%. Moreover, a test system based on a microheater is established for response time measurement, and the response time is measured to be 6.9 ms. Correspondingly, the response time of the same device was 12.7 ms when adopting a traditional chopper—based test system. Therefore, the microheater—based test system improves the measurement accuracy of response time by 46.7%. The result demonstrates that the method of the microheater—based test system is simpler, and has higher accuracy as well as effectiveness. Furthermore, it is an easier way to integrate the microheater into the thermopile device through a CMOS—compatible process. Hence, this test system has greater development potential in the thermal response device, especially in the application with strict requirements on the response time of devices.

## Figures and Tables

**Figure 1 micromachines-13-01717-f001:**
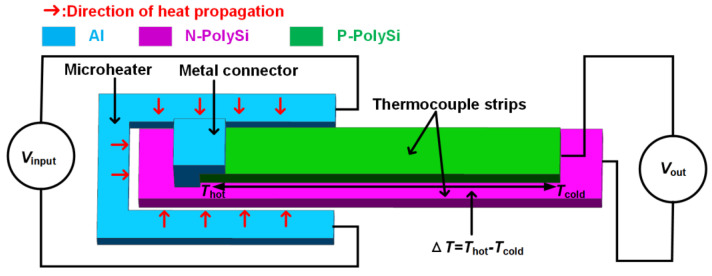
Working principle diagram of a microheater embedded around the hot end of a thermocouple.

**Figure 2 micromachines-13-01717-f002:**
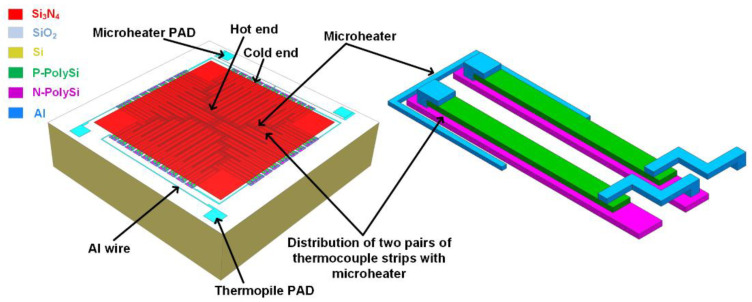
Schematic diagram of a MEMS thermopile with an embedded microheater.

**Figure 3 micromachines-13-01717-f003:**
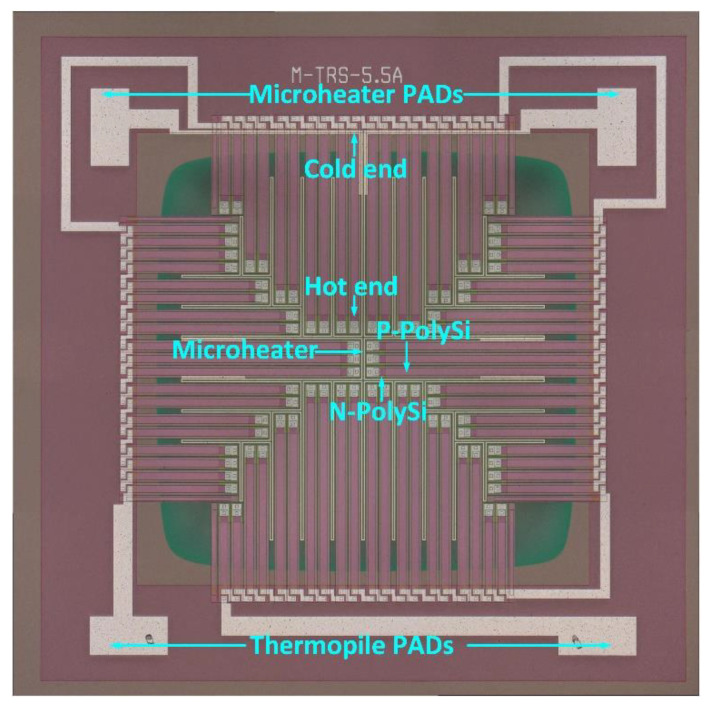
Microscope image of the MEMS thermopile with a microheater.

**Figure 4 micromachines-13-01717-f004:**
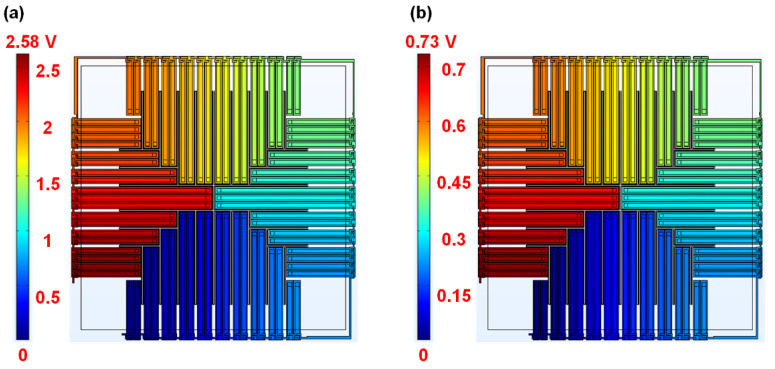
Voltage distribution of thermopile devices embedded with different microheaters: (**a**) Al—based; (**b**) Pt—based.

**Figure 5 micromachines-13-01717-f005:**
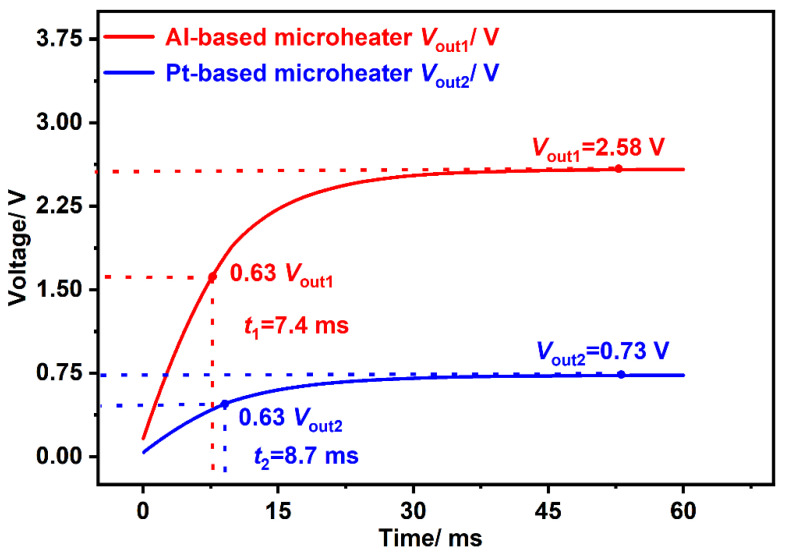
Simulated output voltages and response times of the same device embedded with an Al—based microheater and a Pt—based microheater.

**Figure 6 micromachines-13-01717-f006:**
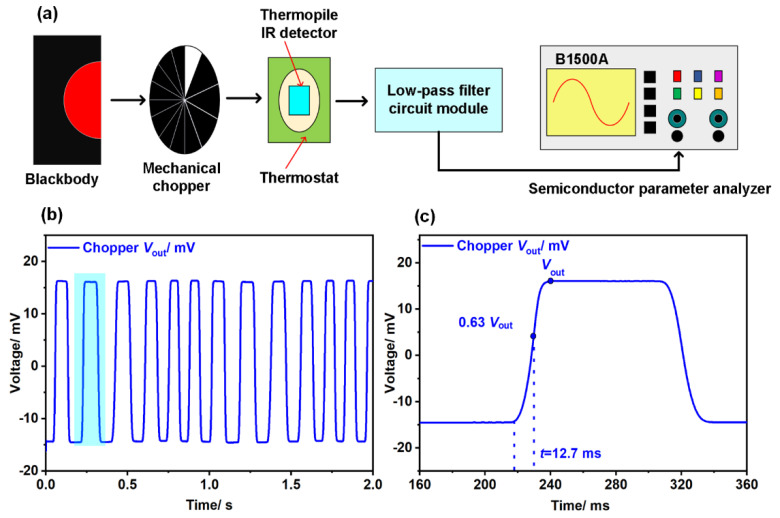
(**a**) Schematic diagram of a chopper—based measurement system; (**b**) output waveform of multiple cycles obtained by the chopper—based system; (**c**) a single cycle demonstrating the response time of the device.

**Figure 7 micromachines-13-01717-f007:**
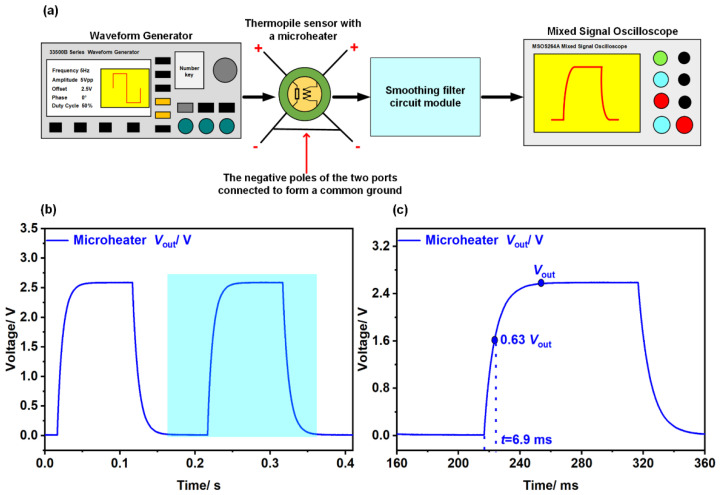
(**a**) Schematic diagram of a microheater—based system; (**b**) output waveform of multiple cycles obtained by the microheater—based system; (**c**) a single cycle illustrating the response time of the device.

**Table 1 micromachines-13-01717-t001:** Key structural parameters of the device.

Parameters	Length (µm)	Width (µm)	Thickness (µm)	Interval (µm)
N—PolySi	300–620	30	0.42	3
P—PolySi	260–580	18	0.42	15
Microheater	/	3	0.5	3

**Table 2 micromachines-13-01717-t002:** Characteristics of materials suitable for microheaters at room temperature [29].

Materials	η (Ω.m)	*m* (10^−4^/K)	ρ (kg/m^3^)	c (J/(kg.K))	λ (W/(m.K))
Al	2.69 × 10^−8^	42.0	2700	904	237
Cu	1.67 × 10^−8^	43.0	8960	384	401
Au	2.30 × 10^−8^	39.0	19,300	129	317
Fe	9.71 × 10^−8^	65.1	7860	449	80.2
Ni	6.84 × 10^−8^	68.1	8900	445	90.7
Pt	10.6 × 10^−8^	39.2	21,450	133	71.6
Ag	1.63 × 10^−8^	41.0	10,500	235	429
W	5.50 × 10^−8^	46.0	19,350	132	174
PolySi	4 × 10^−6^∼1 × 10^−1^	−250∼10	2320	678	31

**Table 3 micromachines-13-01717-t003:** Thermopile sensor response time, measured by different test systems.

Reference	Resistance (KΩ)	Thermocouples	t (ms)	Test Method
20	485.5	N—Poly/P—Poly	14.46	Chopper
22	458.5	N—Poly/P—Poly	14.46	Chopper
25	124.7	N—Poly/Al	16.8	Chopper
26	29	P—Poly/Al	126	Chopper
28	270	Al/P—Poly	10	Chopper
This work	195	N—poly/P—poly	12.7	Chopper
This work	195	N—Poly/P—Poly	6.9	Microheater

## Data Availability

Not applicable.

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
