# Peer review of "A Highly Accurate Method for Measuring Response Time of MEMS Thermopiles"

_micromachines, 2022, doi:10.3390/mi13101717_

Round 1
Reviewer 1 Report
This paper presents a new way to measure the response time of MEMS thermopiles. This may be helpful for providing more efficient and high-accurate results, it seems the time was cut by nearly half compared with the traditional method. I think the topic of this paper is interesting and the organization of the paper is good. English of the paper is mostly fine. Here are some of the issues with this paper that I would like to point out:
1) In line 49, it is said “To avoid such an error, Zhang et al. adopted a pulsed laser-based system for response time measurement.” What’s the precision of the method introduced by Zhang et al. Did you achieve an improvement in measurement accuracy, which should be a key parameter of the thermopiles? As well, the accuracy of current methods that be obtained should be described in the introduction.
2) This paper proposes a measurement method, and an important point is how to measure the accuracy of the method, the accuracy of measurement will directly affect the credibility of the results, and it is necessary to introduce it in detail. Can you give a quantitative analysis of the error?
Reviewer 2 Report
The authors create an efficient method for testing the thermopiles sensor. The method, as simple as it is, is effective as it results from the presented results.
After analyzing international article databases, we identified works on the same topic - A Highly Accurate Method for Measuring Response Time of 2 MEMS Thermopiles.
I ask the authors to include in the article a comparison of their results with other similar works from the specialized literature. Why is their method better? It is necessary to clarify this aspect in order to increase the importance of the presented theme.
Reviewer 3 Report
In this manuscript, the authors develop an approach to accurately measure the response time of MEMS thermopiles through in-situ integration of an Al microheater in the sensor. In my opinion, this manuscript is interesting to the readers of IJMS. The topic is very important in this field. This work is novel and original. The authors have solid background in this field. Therefore, the referee recommends it to be published after the following revisions:
1. The English should be polished by a native speaker.
2. How many different samples of the sensor were created and how good was the reproducibility?
3. Did you perform any optimization for the sensor length, width, thickness, width and interval?
4. How are the performances here compared with state-of-the-art reports? The readers would like to see a paragraph near the end of the manuscript before the conclusion to dedicate to such comparison.

Round 2
Reviewer 2 Report
The authors responded to the requested clarifications.